# Three Year Follow-Up of Reduced Dose of Direct Oral Anticoagulants for Extended Treatment of Venous Thromboembolism: An Ambispective Cohort Study

**DOI:** 10.3390/diagnostics15172283

**Published:** 2025-09-08

**Authors:** Emanuele Valeriani, Arianna Pannunzio, Tommaso Brogi, Ilaria Maria Palumbo, Danilo Menichelli, Silvia Marucci, Luca Tretola, Claudio Maria Mastroianni, Daniele Pastori, Pasquale Pignatelli

**Affiliations:** 1Department of General Surgery and Surgical Specialty, Sapienza University of Rome, Piazzale Aldo Moro 5, 00185 Rome, Italy; arianna.pannunzio@uniroma1.it (A.P.);; 2Department of Infectious Disease, Umberto I Hospital, Viale del Policlinico 155, 00161 Rome, Italy; 3Department of Clinical Internal, Anesthesiological and Cardiovascular Sciences, Sapienza University of Rome, 00185 Rome, Italy; 4IRCCS Neuromed, 86077 Pozzilli, Italy

**Keywords:** anticoagulants, direct-acting oral anticoagulant, pulmonary embolism, venous thromboembolism

## Abstract

**Background:** Few data are available on the outcomes of patients with venous thromboembolism (VTE) on long-term reduced dose of direct oral anticoagulants (DOACs). We evaluated the effectiveness and safety of reduced dose of DOACs for the extended treatment of VTE. **Methods:** In this monocenter, ambispective cohort study, 140 patients receiving a reduced dose of DOACs for VTE were included. The primary outcomes were recurrent VTE, major bleeding and clinically relevant non-major bleeding. The secondary outcomes were arterial events and minor bleedings. The incidence of the primary outcomes was calculated. The rate for secondary outcomes was descriptively reported. **Results:** The mean age of the overall cohort was 72 years. Half of the patients were female, 51.4% had a persistent risk factor, 40.0% an unprovoked VTE, and 8.6% a minor transient risk factor. Most patients had lower extremity deep vein thrombosis with or without pulmonary embolism (55.0%) and received apixaban (73.6%) or rivaroxaban (14.3%) for a mean duration of 2.7 years. Regarding the primary outcomes, there was one recurrent VTE (0.7%), four major bleedings (2.9%) and two clinically relevant non-major bleedings (1.4%). Regarding the secondary outcomes, there were four acute ischemic strokes (2.9%) and two minor bleedings (1.4%). **Conclusions:** Reduced dose of DOACs was associated with a low rate of recurrent VTE and an acceptably low rate of bleeding complications. The rate of arterial events during follow-up suggests the need for an assessment of cardiovascular risk factors in this study population.

## 1. Introduction

Venous thromboembolism (VTE) is one of the leading causes of morbidity and mortality worldwide, with an incidence of one in 1000 people each year [1]. Various risk factors and clinical conditions contribute to the development of venous thromboembolism (VTE), potentially affecting the risk of both initial and recurrent events in different ways [2,3]. The recommended treatment of VTE consists of an initial phase of anticoagulant therapy for at least three months [4]. In specific clinical conditions (e.g., persistent risk factors, unprovoked thrombosis), the risk of recurrences remains high beyond the first three months after the index event and extended anticoagulation should be considered [5]. In certain cases, it may be appropriate to extend anticoagulant therapy following VTE associated with a transient risk factor, particularly in patients with a previous VTE history, or when the event was provoked by a minor transient trigger [6]. The decision to offer extended-phase anticoagulation, however, should consider both the risk of recurrent thrombosis and the risk of bleedings on extended treatment [7]. The availability of reduced doses of direct oral anticoagulants (DOACs) has streamlined the long-term management of patients needing extended anticoagulation [8].

While several studies assessed the efficacy and safety of standard dose of DOACs for VTE treatment [9,10], only two randomized controlled trials have been published reporting data on the beneficial effect of reduced dose of DOACs in the long-term period [11,12]. In these trials only two DOACs (i.e., apixaban and rivaroxaban) have been evaluated and the follow-up for outcomes assessment was with a maximum of 12 months [11,12]. Based on this, rivaroxaban 10 mg od and apixaban 2.5 mg bid are recommended for extended therapy in patients with VTE [13]. However, evidence on long-term outcomes in patients receiving reduced dose of DOACs remains limited [14,15].

To address this gap in knowledge, our study aimed to evaluate the effectiveness and safety of reduced doses of DOACs for the extended phase of VTE treatment beyond 12 months.

## 2. Methods

This study is reported in accordance with the Strengthening the Reporting of Observational Studies in Epidemiology statement for observational studies [16]. Informed consent was obtained from all subjects involved in this study. This study was approved by the local ethic committee of Sapienza University (No. 0073/2022–26/01/2022) and was conducted according to the 1975 Declaration of Helsinki.

### 2.1. Study and Patients’ Characteristics

This is a no-profit, observational, ambispective, monocenter cohort study promoted by the Department of Clinical Internal, Anaesthesiologic and Cardiovascular Sciences of Sapienza University of Rome. Consecutive adult patients (aged ≥ 18 years) who attended the antithrombotic center at Policlinico Umberto I from January 2022 through April 2024 were enrolled. Inclusion criteria were a diagnosis of VTE requiring anticoagulant therapy with a reduced dose of DOACs (i.e., apixaban 2.5 mg every 12 h, dabigatran 110 mg every 12 h, edoxaban 30 mg once daily and rivaroxaban 10 mg once daily) for the extended phase of treatment. Exclusion criteria were administration of full doses of DOACs; anticoagulant therapy different from DOACs, e.g., vitamin k antagonists (VKAs), low-molecular-weight heparin (LMWH); an indication for DOACs therapy other than VTE, e.g., atrial fibrillation; denial of informed consent; any contraindication to the administration of DOACs; and unavailability of relevant data.

At baseline, the following data were collected in an electronic dataset: demographic variables (e.g., age and sex categories), risk factors for VTE development (e.g., personal and family history of VTE and transient or persistent risk factors), type and dosage of DOACs, duration of anticoagulation, concomitant therapies and outcomes of interest.

### 2.2. VTE Risk Factor Classification

Risk factors for recurrent VTE were categorized according to the ISTH classification [17]. Major transient risk factors included prolonged immobilization lasting more than 3 days within the past 3 months due to an acute medical illness, cesarean section and surgery under general anesthesia lasting more than 30 min. Persistent risk factors comprised active cancer and other active non-malignant conditions associated with a significant risk of recurrent VTE after discontinuation of anticoagulant therapy (e.g., inflammatory bowel disease, relevant thrombophilia). Admission to hospital or bed rest out of hospital for less than 3 days for an acute illness, reduced mobility related to leg injury for at least 3 days, surgery with general anesthesia lasting less than 30 min, pregnancy, puerperium or estrogen therapy were considered as minor transient risk factors, all occurring in the past 2 months.

### 2.3. Outcomes

The primary effectiveness outcome included on-treatment recurrent VTE during the follow-up period (i.e., thrombosis in a deep vein of the lower or upper limbs, pulmonary embolism, splanchnic or abdominal vein thrombosis and cerebral vein thrombosis). The secondary effectiveness outcome encompasses on-treatment arterial events (i.e., acute myocardial infarction, acute ischemic stroke, acute limb ischemia).

The primary safety outcome included on-treatment major and clinically relevant non-major bleeding defined by the International Society on Thrombosis and Hemostasis criteria, while the secondary safety outcome included minor bleedings [18,19].

All patients were prospectively followed in our antithrombotic center with follow-up visits scheduled according to the judgement of the attending physician. In case of an outcome occurrence, relevant clinical data were requested from the patients to confirm and further detail the event.

### 2.4. Statistical Analysis

The patients’ baseline characteristics are presented using descriptive statistics. Categorical variables are summarized as counts and percentages. Continuous variables are reported as mean and standard deviation (SD), or as median and interquartile range (IQR), depending on their distribution. Data distribution was assessed using the Shapiro–Wilk test. The patients were categorized into three groups based on their underlying VTE risk factor: unprovoked VTE, persistent risk factor or minor transient risk factor. Additionally, the patients were grouped according to the type of DOACs they received (i.e., apixaban, dabigatran, edoxaban or rivaroxaban). Comparisons of categorical variables were performed using either the chi-squared test or the Fisher’s exact test, while continuous variables were compared using the Student’s *t*-test or the Mann–Whitney U test, as appropriate. The incidence of primary outcomes was expressed in patient-years with corresponding 95% confidence intervals (CIs). All other outcomes were reported using descriptive statistics. Statistical analyses were conducted using RStudio (version 2023.09.1+494, R Core Development Team, Vienna, Austria) [20].

## 3. Results

### 3.1. Baseline Patients’ Characteristics

Table 1 reported the baseline characteristics of included patients. A total of 140 patients were consecutively enrolled. The overall mean age was 72 ± 15 years. The patients with a persistent risk factor were younger (67 ± 16 years) compared to those with unprovoked VTE (76 ± 13 years) and those with a minor transient risk factor (79 ± 5 years). The proportion of females and the mean BMI were similar across the three subgroups. The proportion of patients with a history of previous VTE or a family history of VTE was higher among those with a persistent risk factor (31.9% and 5.5%, respectively) and those with unprovoked VTE (44.6% and 8.9%) compared to the patients with a minor transient risk factor (8.3% and 0%).

Overall, most patients (55.0%) had lower extremity deep vein thrombosis with or without pulmonary embolism, followed by isolated pulmonary embolism (30.7%). Few patients had upper extremity deep vein thrombosis with or without pulmonary embolism (5.1%) and atypical site thrombosis (9.2%). There were no relevant differences across subgroups of patients in terms of the site of thrombosis.

In the subgroup of patients with persistent risk factors, the most frequent one was thrombophilia (40.3%), followed by active cancer (30.6%) and autoimmune disease (19.4%). One patient (1.4%) had a liver cirrhosis-related splanchnic vein thrombosis while six patients (8.3%) had multiple persistent risk factors. In the subgroup of patients with minor transient risk factors, 50.0% had an acute illness, 41.7% a surgery and 8.3% a leg injury.

### 3.2. Type of DOACs and Duration of Anticoagulation

Overall, 73.6% of patients were treated with apixaban, while 14.3% received rivaroxaban. A smaller proportion received edoxaban (6.4%) or dabigatran (5.7%). There were no significant differences in the type of DOAC prescribed across patient subgroups. In contrast, the mean duration of treatment was longer in patients with a persistent risk factor (2.8 ± 2.0 years) and in those with unprovoked VTE (3.0 ± 2.2 years), compared to patients with a minor transient risk factor (1.6 ± 1.5 years). Baseline characteristics according to the type of DOACs are reported in Appendix A.

Overall, 4.3% of patients received concomitant antiplatelets, 27.9% concomitant statins and 13.7% concomitant steroid therapy.

### 3.3. Primary Outcomes

The incidence of the primary outcomes is reported in Table 2 and Table 3.

Overall, one case of recurrent lower extremity deep vein thrombosis (0.7%) was observed. It occurred after 16 months of apixaban therapy in a patient with significant thrombophilia (heterozygous for both factor V Leiden and prothrombin gene mutation). This patient was started on a full dose of apixaban and did not develop any other thrombotic or bleeding events during the remaining 9 months of follow-up (Table 4).

Four major bleedings (2.9%) occurred during follow-up. Of these bleedings, one occurred in the gastro-intestinal tract after 15 months, one in the genitourinary tract after 23 months, one was cutaneous after 76 months, and one was an epistaxis after 24 months of therapy. All major bleedings occurred during apixaban administration. A reduced dose of apixaban was resumed in two patients after 6 days and 59 days from suspension, and the remainder of the follow-up period was without any other events. No recurrent VTE or arterial events occurred in the two patients who remained without anticoagulant therapy (Table 4).

There were two clinically relevant non-major bleedings (1.4%) during follow-up. One of these occurred in the gastro-intestinal tract after 36 months of apixaban; no recurrent VTE or arterial events occurred during the 13 months of follow-up after anticoagulant therapy was discontinued. The other bleeding occurred in the genitourinary tract after 2 months of apixaban; anticoagulation was not discontinued, and no further events occurred during the following 58 months of follow-up (Table 4).

Table 4 reports the characteristics of patients who developed the outcome of interest.

### 3.4. Secondary Outcomes

Overall, four acute ischemic strokes (2.9%) occurred after 7, 10, 53 and 56 months of anticoagulation. Three of these events occurred during apixaban and one during dabigatran therapy. Aspirin therapy was initiated in one patient with active cancer, and low-molecular-weight heparin was initiated instead of apixaban in one patient with unprovoked thrombosis.

Of the two minor bleedings (1.4%), one occurred in the genitourinary tract after two months of apixaban, and the other was cutaneous and occurred after 37 months of edoxaban. No patient discontinued anticoagulant therapy.

Table 4 reports the characteristics of patients who developed the outcome of interest.

## 4. Discussion

Our study, including VTE patients with up to 3 years of follow-up, shows a low incidence of recurrent VTE and major bleeding during treatment with reduced dose of DOACs. Of note, the incidence of arterial thrombotic events exceeded that of recurrent VTE.

The available randomized trials provided some useful information for the management of these patients. More specifically, the administration of a reduced dose of apixaban, compared to the full dose, was associated with a similar risk of recurrent VTE (1.7% versus 1.7%; hazard ratio: 0.97, 95% CI: 0.46 to 2.02) and a trend toward reduced risk of bleeding complications (3.2% versus 4.3%; hazard ratio: 0.74, 95% CI: 0.46 to 1.22) in the AMPLIFY-EXT trial [11]. The results were similar in the EINSTEIN CHOICE trial, showing a similar efficacy in term of recurrent VTE (1.2% versus 1.5%; hazard ratio: 1.34, 95% CI: 0.65 to 2.75) and a trend toward better safety in term of bleeding complications (2.4% versus 3.3%; hazard ratio: 1.37, 95% CI: 0.83 to 2.26) with the reduced dose compared to the full dose of rivaroxaban [12].

Considering the risk of recurrent VTE as maximum in the first year after the index event, the results of these two trials are extremely useful [5,11,12]. It should be acknowledged, however, that roughly 60% of patients reached the 12 months of treatment and that most of the included patients had an unprovoked VTE and thus are not completely representative of the real-world population requiring these therapies [11,12]. A further limitation is that their follow-up period is not sufficient to address the management of reduced therapy over a longer time frame. The risk of recurrent VTE, indeed, persists beyond the first year after the index unprovoked or provoked event in a non-negligible proportion of patients who discontinued anticoagulation, thereby questioning the benefit of lifelong therapy [5,21,22].

In this context, the present study extends previous findings by providing data from a longer follow-up period (2.7 ± 2.1 years), showing that the rate of recurrent VTE is low, and that of bleeding complications is acceptably low [10,11,12]. It should be noted that the rate of recurrent VTE observed in our study, although lower than that reported in previous RCTs, is comparable to recent real-world data—including patients from an Italian registry—and highlights the importance of follow-up in specialized antithrombotic centers [15]. Even if it goes beyond the objective of the present study, the lower incidence of recurrent VTE and the higher incidence of major bleeding compared to previous reports may suggest that the decision to extend treatment with reduced dose of DOACs beyond the first 12 months should mostly be based on the patient’s individual bleeding risk [10,11,12].

A further interesting finding of our study is that 2.9% of patients, most of whom has persistent risk factors, experienced an arterial event during follow-up. This rate is higher than previously reported, highlighting the urgent need of a multidisciplinary approach, with particular attention to the prevention of primary arterial events during long-term follow-up in patients with VTE [23]. This result highlights the need for a more comprehensive assessment of both modifiable and non-modifiable cardiometabolic risk factors, as well as the potential benefits of concomitant statin or antiplatelet therapy in addition to anticoagulation. Specifically, statin treatment may be an adjunctive therapy in addition to anticoagulation that may be considered in these patients, as it is associated with a reduction in VTE risk, along with a well-established reduction in atherothrombotic events [24,25,26]. While concomitant antiplatelet therapy does not appear to significantly impact the risk of recurrent VTE, it may help reduce overall cardiovascular risk, albeit with an increased risk of bleeding complications [27]. Additionally, it is important to note that reduced doses of DOACs may not provide as strong protection against arterial thrombosis as they do for the secondary prevention of VTE.

### Strength and Limitations

Our study is the first one exploring the efficacy of reduced dose of DOACs beyond the first 12 months of therapy in patients with VTE. This is a setting with few clinical, real-life data, and for this reason, current guidelines and consensus rely on results from available randomized trials with a shorter follow-up period.

Our study has some limitations that warrant discussion. First, it is a monocenter ambispective study which included mostly Caucasian patients, and it is therefore not fully representative of the broader VTE patient population, raising the risk of selection bias. Secondly, the selection and duration of anticoagulant therapy were determined by the treating physician, which may result in variability in treatment protocols. This heterogeneity limited the ability to draw definitive conclusions regarding the effectiveness and safety of anticoagulation and precluded indirect comparisons between different DOACs. It should be acknowledged that most patients received apixaban or rivaroxaban, with little data for dabigatran and edoxaban, further limiting the generalizability of the results. Furthermore, the criteria for dose reduction of dabigatran and edoxaban in the extended treatment of VTE were not well established by current evidence and are not actually recommended for the extended treatment of VTE due to the lack of data. Third, the low number of events did not allow any evaluation of potential risk factors for thrombotic and bleeding events during follow-up. Although the baseline patients’ characteristics did not significantly differ among patients with different VTE risk factors, we cannot exclude residual confounding related to unmeasured variables that possibly affected the occurrence of the outcomes. Furthermore, the nature of this study may have increased the risk of missing outcome events.

In conclusion, reduced dose of DOACs were associated with a low rate of recurrent VTE and an acceptably low rate of bleeding complications during a very long-term follow-up period. The rate of arterial events during the follow-up period suggests the need for proactive identification and management of cardiovascular risk factors.

## Figures and Tables

**Table 1 diagnostics-15-02283-t001:** Characteristics of overall population and according to persistent risk factors, transient RF and unprovoked VTE.

Variables	Overall n = 140	Persistent RFs n = 72	Unprovoked n = 56	Minor Transient RFs n = 12	*p*-Value
Risk factors					
Mean age, years (SD)	72 (±15)	67 (±16)	76 (±13)	79 (±5)	0.003
Female sex, n (%)	73 (52.1)	37 (51.4)	30 (53.6)	6 (50.0)	0.959
Mean BMI, kg/m^2^ (SD)	27 (±5.8)	27 (±6)	29 (±6)	26 (±2)	0.279
Previous cancer, n (%)	22 (15.1)	15 (20.8)	7 (12.5)	0	0.114
Previous VTE, n (%)	49 (35.0)	23 (31.9)	25 (44.6)	1 (8.3)	0.047
Family history of VTE, n (%)	9 (5.3)	4 (5.5)	5 (8.9)	0	0.429
Previous bleeding, n (%)	5 (3.6)	2 (2.8)	1 (1.8)	2 (16.7)	0.040
Site of VTE					0.056
PE, n (%)	43 (30.7)	26 (36.1)	15 (26.8)	2 (16.7)	
LEDVT, n (%)	51 (36.4)	25 (34.7)	23 (41.1)	3 (25.0)	
LEDVT + PE, n (%)	26 (18.6)	10 (13.9)	11 (19.5)	5 (41.7)	
UEDVT, n (%)	5 (3.6)	5 (6.9)	0	0	
UEDVT + PE, n (%)	2 (1.5)	1 (1.4)	1 (1.8)	0	
SplVT, n (%)	3 (2.1)	3 (4.2)	0	0	
Retinal vein occlusion, n (%)	1 (0.7)	1 (1.4)	0	0	
Cerebral vein thrombosis, n (%)	1 (0.7)	0	1 (1.8)	0	
Jugular vein thrombosis, n (%)	4 (2.9)	0	3 (5.4)	1 (8.3)	
Ovarian vein thrombosis, n (%)	1 (0.7)	0	0	1 (8.3)	
Thrombosis at other sites, n (%)	3 (2.1)	1 (1.4)	2 (3.6)	0	
Cause of VTE					<0.001
Thrombophilia, n (%)	29 (20.7)	29 (40.3)	0	0	
Active cancer, n (%)	22 (15.7)	22 (30.6)	0	0	
Autoimmune disease, n (%)	14 (10.0)	14 (19.4)	0	0	
Liver cirrhosis, n (%)	1 (0.7)	1 (1.4)	0	0	
Multiple persistent RFs, n (%)	6 (4.3)	6 (8.3)	0	0	
Acute illness, n (%)	6 (4.3)	0	0	6 (50.0)	
Surgery, n (%)	5 (3.6)	0	0	5 (41.7)	
Leg injury, n (%)	1 (0.7)	0	0	1 (8.3)	
Therapy					0.260
Type of anticoagulant, n (%)					
Apixaban, n (%)	103 (73.6)	55 (76.4)	41 (73.2)	7 (58.3)	
Dabigatran, n (%)	8 (5.7)	5 (6.9)	3 (5.4)	0	
Edoxaban, n (%)	9 (6.4)	3 (4.2)	3 (5.4)	3 (25.0)	
Rivaroxaban, n (%)	20 (14.3)	9 (12.5)	9 (16.0)	2 (16.7)	
Treatment length, years (SD)	2.7 (2.1)	2.8 (2.0)	3.0 (2.2)	1.6 (1.5)	0.096
Concomitant antiplatelet, n (%)	6 (4.3)	2 (2.8)	4 (7.1)	0	0.359
Concomitant statin, n (%)	39 (27.9)	21 (29.2)	15 (26.8)	3 (25.0)	0.931
Concomitant steroid, n (%)	19 (13.7)	15 (20.8)	4 (7.1)	0	0.026

BMI: body mass index; CVC: central venous catheter; LEDVT: lower extremity deep vein thrombosis; PE: pulmonary embolism; RFs: risk factors; SD: standard deviation; SplVT: splanchnic venous thrombosis; UEDVT: upper extremity deep vein thrombosis; VTE: venous thromboembolism.

**Table 2 diagnostics-15-02283-t002:** Outcomes assessed for overall population and stratified by persistent risk factors, transient RFs and unprovoked VTE.

Outcomes	Overall n = 140	Persistent RFs n = 72	Unprovoked n = 56	Minor Transient RFs n = 12	*p*-Value
Recurrent VTE, n (%)	1 (0.7)	1 (1.4)	0 (0.0)	0	0.621
Arterial events, n (%)	4 (2.9)	3 (4.2)	1 (1.8)	0	0.598
Bleeding					0.310
Major, n (%)	4 (2.9)	3 (4.2)	1 (1.8)	0	
CRNMB, n (%)	2 (1.4)	0	2 (3.6)	0	
Minor, n (%)	3 (2.1)	0	2 (3.6)	0	

CRNMB: clinically relevant non-major bleeding; RFs: risk factors; VTE: venous thromboembolism.

**Table 3 diagnostics-15-02283-t003:** Incidence rates of the primary effectiveness and safety outcomes.

Outcome	Incidence × 1000 Patients/Year (95% CI)
Recurrent VTE	3 (1 to 14)
Major bleedings	10 (3 to 26)
Clinically relevant non-major bleeding	5 (1 to 20)

CI, confidence intervals; VTE, venous thromboembolism.

**Table 4 diagnostics-15-02283-t004:** Characteristics of patients who developed a primary or secondary outcome.

Index Event	Outcome	Risk Factors for the Event	DOAC	Months Since DOAC Start	Treatment Modification	Occurrence of Other Outcomes After Treatment Modification	Residual Follow-Up Months
VTE recurrence							
LEDVT	LEDVT	Thrombophilia	Apixaban 2.5 mg bid	16	Apixaban 5 mg bid	None	54
Arterial events							
UEDVT	Acute ischemic stroke	Active cancer, diabetes mellitus, arterial hypertension, dyslipidemia, carotid atherosclerosis	Apixaban 2.5 mg bid	53	Aspirin and apixaban	None	24
LEDVT	Acute ischemic stroke	Hypertension, psoriatic arthritis, thrombophilia	Apixaban 2.5 mg bid	10	Switch to LMWH, then apixaban 5 mg bid	None	29
LEDVT	Acute ischemic stroke	Active cancer, hypertension, previous cardiovascular events, thrombophilia	Dabigatran 110 mg bid	7	None	None	19
LEDVT	Acute ischemic stroke	Thrombophilia, arterial hypertension	Apixaban 2.5 mg bid	56	None	None	48
Major bleedings							
PE	Gastro-intestinal	Chron disease	Apixaban 2.5 mg bid	15	Discontinuation of anticoagulant treatment	None	12
PE	Genito-urinary	None	Apixaban 2.5 mg bid	23	Discontinuation of nticoagulant treatment and resumption after 59 days	None	14
PE	Cutaneous	None	Apixaban 2.5 mg bid	76	Discontinuation of anticoagulant treatment and resumption after 6 days	None	3
Retinal vein thrombosis	Epistaxis	None	Apixaban 2.5 mg bid	24	Discontinuation of anticoagulant treatment	None	36
CRNMB							
LEDVT	Gastro-intestinal	None	Apixaban 2.5 mg bid	36	Discontinuation of anticoagulant treatment	None	13
LEDVT	Genito-urinary	History of prostatic cancer	Apixaban 2.5 mg bid	2	None	None	58
Minor bleedings							
PE	Genito-urinary	None	Apixaban 2.5 mg bid	2	None	None	7
LEDVT + PE	Cutaneous	None	Edoxaban 30 mg	37	None	None	18

CRNMB: clinically relevant non-major bleeding; LEDVT: lower extremity deep vein thrombosis; LMWH: low molecular weight heparin; OAT: oral anticoagulant treatment; TIA: transient ischemic attack; VTE: venous thromboembolism.

## Data Availability

All data are available upon request at emanuele.valeriani@uniroma1.it.

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
