# Peer review of "Three Year Follow-Up of Reduced Dose of Direct Oral Anticoagulants for Extended Treatment of Venous Thromboembolism: An Ambispective Cohort Study"

_diagnostics, 2025, doi:10.3390/diagnostics15172283_

Round 1
Reviewer 1 Report
Comments and Suggestions for Authors
The study addresses an important clinical question — the long-term effectiveness and safety of reduced-dose DOACs for extended VTE treatment — an area with limited real-world data beyond 12 months.
Here, some minor suggestions:
- Add some recent relevant works in the introduction and discussion, such as: Couturaud F, et al. Extended treatment of venous thromboembolism with reduced-dose versus full-dose direct oral anticoagulants in patients at high risk of recurrence: a non-inferiority, multicentre, randomised, open-label, blinded endpoint trial. Lancet. 2025;405(10480):725-735.
- Explicitly state how recurrent VTE and bleeding events were monitored—whether via active follow-up, scheduled imaging, or passive reporting.
- The unexpected higher incidence of arterial events (~2.9%) warrants a more in-depth discussion on cardiometabolic risk, potential statin or antiplatelet optimization, and the emerging evidence that reduced DOAC dose may not mitigate arterial thrombosis risk as strongly as it does VTE
Author Response
Point 1: The study addresses an important clinical question — the long-term effectiveness and safety of reduced-dose DOACs for extended VTE treatment — an area with limited real-world data beyond 12 months.
Here, some minor suggestions:
Add some recent relevant works in the introduction and discussion, such as: Couturaud F, et al. Extended treatment of venous thromboembolism with reduced-dose versus full-dose direct oral anticoagulants in patients at high risk of recurrence: a non-inferiority, multicentre, randomised, open-label, blinded endpoint trial. Lancet. 2025;405(10480):725-735.
Reply: We thank the Reviewer for the positive feedback and insightful comments. According to this comment, we added recent relevant works (DOI: 10.1016/j.jtha.2025.07.034, DOI: 10.1016/S0140-6736(24)02842-3) through Introduction and discussion.
Point 2: Explicitly state how recurrent VTE and bleeding events were monitored—whether via active follow-up, scheduled imaging, or passive reporting.
Reply: According to this comment we added the following sentence in Methods (page 7, rows 107 to 110): “All patients were prospectively followed in our antithrombotic center with follow-up visit scheduled according to the judgement of the attending physician. In case of an outcome occurrence, relevant clinical data were requested from the patients to confirm and further detail the event.”
Point 3: The unexpected higher incidence of arterial events (~2.9%) warrants a more in-depth discussion on cardiometabolic risk, potential statin or antiplatelet optimization, and the emerging evidence that reduced DOAC dose may not mitigate arterial thrombosis risk as strongly as it does VTE
Reply: According to this comment we added the following sentences in Discussion (page 8, row 302 and page 9 rows 304 to 314): “This result highlights the need for a more comprehensive assessment of both modifiable and non-modifiable cardiometabolic risk factors, as well as the potential benefits of concomitant statin or antiplatelet therapy on top of anticoagulation. Specifically, statin treatment may be an adjunctive therapy on top of anticoagulation that may be considered in these patients as it is associated with a reduction of VTE risk along with a well-established reduction of atherothrombotic events [24-26]. While concomitant antiplatelet therapy does not appear to significantly impact the risk of recurrent VTE, it may help reduce overall cardiovascular risk, albeit with an in-creased risk of bleeding complications.[27] Additionally, it is important to note that re-duced doses of DOACs may not provide as strong protection against arterial thrombosis as they do for secondary prevention of VTE.”
Reviewer 2 Report
Comments and Suggestions for Authors
Dear Authors! Thank you for the interesting data presented in your paper. I believe it is important to discuss extended anticoagulation beyond the first 12 months in VTE patients.
I have two major remarks that have to be addressed and discussed.
- You registered low rate of VTE recurrence which was 0,7%. You have mentioned three studies with similar data in a way that makes reader think that your findings are at the same range. But, it’s not like that, on my opinion. The recurrence rate you have found is remarkably low in comparison with those studies. Two of the cited studies were randomized with recurrence rates 1.2 to 1.7%. Your real-world findings twice as lower. But. It is generally accepted that real-world rates of VTE recurrence are higher, not lower. In support we can look at the third study that you cited. In a real-world study on extended treatment with edoxaban VTE recurrence rate was 3.8%. So, your rate of 0.7% can be considered as too low. I believe, you have to discuss this in the Discussion section of the paper that way. And the possible reason for this discrepancy has to be discussed.
- The second major remark is related to the first. May be low recurrence rate is connected to difficulties to follow the patients for a long time which is general problem in a real-world settings? The follow up of the patients is not described in details. When the patients were followed after inclusion? Every half year? Every year? Other ways? How did you register recurrent VTE events? Did you just check electronic database? Or did the patient ask for medical help due to symptoms developed? Did the patient with recurrent event visit your center? Did you confirm recurrence based on just a symptoms and sign? Or you did perform duplex ultrasound of leg veins?
Minor remarks
You said that Table 1 is for baseline characteristics of the study group. But, you have also included there data on outcomes. Recurrence and bleedings rates have to be presented separately. It’s better to include these rates in the Table 2.
Author Response
Point 1 and 2: Dear Authors! Thank you for the interesting data presented in your paper. I believe it is important to discuss extended anticoagulation beyond the first 12 months in VTE patients. I have two major remarks that have to be addressed and discussed. You registered low rate of VTE recurrence which was 0,7%. You have mentioned three studies with similar data in a way that makes reader think that your findings are at the same range. But, it’s not like that, on my opinion. The recurrence rate you have found is remarkably low in comparison with those studies. Two of the cited studies were randomized with recurrence rates 1.2 to 1.7%. Your real-world findings twice as lower. But. It is generally accepted that real-world rates of VTE recurrence are higher, not lower. In support we can look at the third study that you cited. In a real-world study on extended treatment with edoxaban VTE recurrence rate was 3.8%. So, your rate of 0.7% can be considered as too low. I believe, you have to discuss this in the Discussion section of the paper that way. And the possible reason for this discrepancy has to be discussed. The second major remark is related to the first. May be low recurrence rate is connected to difficulties to follow the patients for a long time which is general problem in a real-world settings? The follow up of the patients is not described in details. When the patients were followed after inclusion? Every half year? Every year? Other ways? How did you register recurrent VTE events? Did you just check electronic database? Or did the patient ask for medical help due to symptoms developed? Did the patient with recurrent event visit your center? Did you confirm recurrence based on just a symptoms and sign? Or you did perform duplex ultrasound of leg veins?
Reply: We thank the Reviewer for the positive feedback and insightful comments. We acknowledged that our rate of recurrent VTE was lower than previously reported. However, a recent study form START2 registry (DOI: 10.1016/j.jtha.2025.07.034) reported similar incidence of recurrent VTE. So, to address the points:
- We added the following sentences of discussion (page 8, rows 290 to 293) “It should be noted that the rate of recurrent VTE observed in our study, although lower than that reported in previous RCTs, is comparable to recent real-world data—including patients from an Italian registry—and highlights the importance of follow-up in specialized antithrombotic centers.[15]”
- We added the following sentence of Methods (page 7, rows 107 to 110): “All patients were prospectively followed in our antithrombotic center with follow-up visit scheduled according to the judgement of the attending physician. In case of an outcome occurrence, relevant clinical data were requested from the patients to confirm and further detail the event.”
Point 3: Minor remarks. You said that Table 1 is for baseline characteristics of the study group. But, you have also included there data on outcomes. Recurrence and bleedings rates have to be presented separately. It’s better to include these rates in the Table 2.
Reply: We modified Tables according to this comment.
Round 2
Reviewer 2 Report
Comments and Suggestions for Authors
Dear Authors! I'm Ok with your responses to my remarks. I support the paper.
Author Response
Dear Reviewer,
Thanks for your help in improving our work.